# Functional Characterization of POFUT1 Variants Associated with Colorectal Cancer

**DOI:** 10.3390/cancers12061430

**Published:** 2020-05-31

**Authors:** Marlène Deschuyter, Florian Pennarubia, Emilie Pinault, Sébastien Legardinier, Abderrahman Maftah

**Affiliations:** 1PEIRENE, EA 7500, Glycosylation and Cell Differentiation, Faculty of Sciences and Technology, University of Limoges, F-87060 Limoges, France; marlene.deschuyter@unilim.fr (M.D.); florian.pennarubia@uga.edu (F.P.); emilie.pinault@unilim.fr (E.P.); sebastien.legardinier@unilim.fr (S.L.); 2Complex Carbohydrate Research Center, University of Georgia, Athens, GA 30602, USA; 3BISCEm US042 INSERM—UMS 2015 CNRS, Mass Spectrometry Platform, Faculty of Medicine and Pharmacy, University of Limoges, F-87025 Limoges, France

**Keywords:** colorectal cancer, EGF-like domain, mutations, POFUT1

## Abstract

Background: Protein *O*-fucosyltransferase 1 (POFUT1) overexpression, which is observed in many cancers such as colorectal cancer (CRC), leads to a NOTCH signaling dysregulation associated with the tumoral process. In rare CRC cases, with no *POFUT1* overexpression, seven missense mutations were found in human POFUT1. Methods: Recombinant secreted forms of human WT POFUT1 and its seven mutated counterparts were produced and purified. Their *O*-fucosyltransferase activities were assayed in vitro using a chemo-enzymatic approach with azido-labeled GDP-fucose as a donor substrate and NOTCH1 EGF-LD26, produced in *E. coli* periplasm, as a relevant acceptor substrate. Targeted mass spectrometry (MS) was carried out to quantify the *O*-fucosyltransferase ability of all POFUT1 proteins. Findings: MS analyses showed a significantly higher *O*-fucosyltransferase activity of six POFUT1 variants (R43H, Y73C, T115A, I343V, D348N, and R364W) compared to WT POFUT1. Interpretation: This study provides insights on the possible involvement of these seven missense mutations in colorectal tumors. The hyperactive forms could lead to an increased *O*-fucosylation of POFUT1 protein targets such as NOTCH receptors in CRC patients, thereby leading to a NOTCH signaling dysregulation. It is the first demonstration of gain-of-function mutations for this crucial glycosyltransferase, modulating NOTCH activity, as well as that of other potential glycoproteins.

## 1. Introduction

Colorectal cancer (CRC) is the third most diagnosed cancer worldwide (second in females and third in males), with 1.8 million cases in 2018 and 880,792 deaths according to the World Health Organization. CRC is characterized by heterogeneous solid tumors, caused by the accumulation of numerous genetic alterations within cells, as well as epigenetic ones. Among the main causes identified in the progression of benign adenoma to malignant carcinoma, chromosomal instability was incriminated in more than 80% of sporadic cases of CRC [1]. The long arm of chromosome 20 was shown as a frequently dysregulated unstable region in several cancers such as CRC [2]. The region 20q11.21 was found to contain only four genes, including pleomorphic adenomagene-like 2 (*PLAGL2)* and protein *O*-fucosyltransferase 1 (*POFUT1)* [3]. A recent study showed that these two genes (*POFUT1* and *PLAG2*) share a bidirectional promoter and, associated with a copy number amplification, promote colorectal cancer through dysregulation of both Notch and Wnt/ß-catenin signaling pathways [4,5]. The increased proliferation in CRC was attributed to PLAGL2 overexpression impacting the Wnt/ß-catenin pathway [4], while the colorectal tumor progression was correlated to dysregulation of Notch signaling caused by *POFUT1* overexpression [6,7].

POFUT1 is an endoplasmic reticulum (ER)-resident glycosyltransferase allowing *O*-fucosylation of the membrane and secreted glycoproteins within their EGF-like domains (EGF-LDs). This 388 amino-acid-long enzyme (EC 2.4.1.221) in humans, which comprises four disulfide bridges and a terminal KDEL-like ER-retention sequence (RDEF), is an invertase with a GT-B 3D structure characterized by two Rossmann-fold domains [8]. The interface between these two conserved domains forms the catalytic center comprising conserved residues interacting with the GDP-fucose donor substrate [9], and also with acceptor substrates, namely h-type EGF-LDs (hEGF-LDs) [10]. Interestingly, to be soluble and fully functional, POFUT1 must be modified by *N*-glycosylation at its two conserved consensus sites, as was shown for bovine POFUT1 [11]. POFUT1-mediated *O*-fucosylation is a rare post-translational modification occurring in ER, which leads to fucose transfer to serine or threonine residues of hEGF-LDs within the consensus sequence C^2^-XXXX-(S/T) -C^3^, where C^2^ and C^3^ are the second and third conserved cysteines. Among the potentially POFUT1-modified proteins, the most documented were Notch receptors which are extensively modified with *O*-fucoses in their extracellular domains [12,13,14]. Interestingly, Notch receptors *O*-fucosylation was widely reported to be involved in the modulation of their interaction with ligands, thereby controlling the activation of the Notch signaling pathway [15]. Consequently, *O*-fucosylation was shown to be required for regulation of Notch signaling in several physiologic processes, and its alteration can induce pathological situations [16]. Indeed, the knockout of *Pofut1* in mice is lethal at midgestation [17], with embryos displaying the same severe phenotype as that of KO mice for Notch1 or *Rbp-jk*, the main transcriptional repressor of Notch signaling [18,19]. POFUT1 overexpression was observed in many cancers affecting different organs such as the liver [20], stomach [21], oral cavity [22], or breast [23]. The increased quantity of POFUT1 in hepatocellular carcinomas was associated with a poor prognosis. This was due to an abnormal NOTCH activation that led to in vitro increase of both proliferation and migration [20]. More recently, *POFUT1* overexpression, mainly due to a 20q11.21 amplification and a subsequent increase of the *POFUT1* copy number, was also detected in CRC from the stage I leading to a dysregulation of Notch signaling [6]. Remarkably, the previous study showed that among tumors with amplification of the unstable 20q.11.21 region (76%), 90% of tumors from CRC patients exhibited *POFUT1* overexpression. However, in less frequent CRC cases (19%), with no chromosomal amplification and no change in POFUT1 quantity, seven missense mutations were found for POFUT1 with no consequences demonstrated on its enzymatic activity. These seven point-mutations (R43H, Y73C, T115A, S300L, I343V, D348N, and R364W), found in CRC using databases reporting single-nucleotide mutations in cancers, were predicted to be associated or not with a malignant prognosis.

Given the importance of the *O*-fucosylation in the regulation of Notch receptors-ligands interactions and its consequences on Notch signaling activation, we chose to investigate the functional significance of these seven POFUT1 variants found in CRC. For this purpose, we first produced and purified soluble forms of WT and mutated POFUT1 variants in stable CHO cell lines and a correctly folded and non-glycosylated EGF-LD, as a substrate, in the bacterial periplasm. More precisely, the ability of fucose transfer was determined in vitro for each POFUT1 variant using GDP-fucose (azido-labeled or not) and NOTCH1 EGF-LD 26, known to be modified on its highly conserved *O*-fucose site [24]. After in vitro *O*-fucosylation reactions, copper-catalyzed azide-alkyne cycloaddition (CuAAC) referred to as click chemistry [25], and multiple reaction monitoring-mass spectrometry (MRM-MS), were performed, as previously described in Reference [25]. We did this provide evidence and quantify the *O*-fucose transfer activity of each POFUT1 variant compared to the WT one. The prediction of GDP-fucose binding to a mutated POFUT1 variant (R43H) by automatic molecular docking, as well as the current knowledge about interactions of POFUT1 with its donor and acceptor substrates, allowed us to discuss the impact of mutations on activities of POFUT1 variants, especially those with mutations close to the GDP-fucose binding cavity. In addition to investigations about the functional state of mutated POFUT1 proteins carried by CRC patients, this study also provides additional information on the structure–function relationships of this glycosyltransferase.

## 2. Results

### 2.1. Frequency and Location of Missense Mutations Affecting POFUT1 in CRC

According to the BioMuta database [26,27,28], reporting non-synonymous single-nucleotide mutations associated with cancers, the mutations R43H, Y73C, and T115A were found twice in POFUT1 of CRC patients, while the others (S300L, I343V, D348N, R364W) were observed once. Interestingly, one CRC patient was found to carry both mutations R43H and Y73C (TCGA-D5-6540-01) using the Firebrowse database. Our study could be helpful to speculate about the potential cumulative or compensatory effect of these two mutations on POFUT1 activity in this patient.

We aligned peptide sequences encoding POFUT1 of several species (Figure 1). The human sequence, which is very similar to that of the mouse, exhibited many differences to nematode *C. elegans*. However, alignments showed several conserved regions, where some were known to be involved in binding to donor substrate, namely the regions R43-N46, H238-R240, and S357-F358 [29]. Interestingly, the residue R^43^, mutated to histidine in CRC (R43H), and located in the first substrate-binding region, was known to directly interact with the fucose moiety of the donor substrate (GDP-fucose) [30]. This crucial region also contains three highly conserved residues, namely M^41^, G^42^, and N^46^ in humans, known to establish important links through sulphur-hydrogen or hydrogen bonds with the C^2^-C^3^ subdomain of hEGF-LDs from mouse proteins [10]. The highly conserved Y^73^, mutated to cysteine in CRC (Y73C), was also demonstrated to be involved in binding to hEGF-LD [10]. For the three other highly conserved residues mutated in CRC (I343V, D348N, and R364W) and located close to the third substrate-binding region S357-F358, their contribution to POFUT1 activity was unknown yet. On the contrary, the two last mutations found in CRC, namely S300L and T115A, were considered to be non-conserved residues among species. Overall, the high conservation of at least five residues, among the seven mutated ones in CRC patients, as well as the implication of three of them in other types of cancers suggested functional consequences on POFUT1 activity.

### 2.2. Production of Soluble Forms for Human WT and Mutated POFUT1 Variants

When structure-function studies should be performed on POFUT1, major drawbacks related to the isolation of this glycosyltransferase should be considered due to its status of ER-resident protein. To overcome this issue, we produced recombinant secreted proteins for human WT and mutated POFUT1 variants after establishing stable transfected CHO Flp-In^TM^ cell lines. For this purpose, a modified expression vector named pSec-NtermHis6 [25] was used to allow secretion of recombinant POFUT1, with an N-terminal polyhistidine tag and devoided of its C-terminal ER-retention signal RDEF.

To verify secretion of all recombinant POFUT1 variants, we performed western blot analyses with equal volumes of supernatants from CHO conditioned culture media using a specific anti-POFUT1 antibody. As shown in Figure 2, recombinant proteins were secreted unequally. Using medium supplemented with 10% Fetal Bovine Serum (FBS), only the secreted variants I343V and R364W were weakly detected in the supernatant. However, strong signals of equivalent intensity were seen for the other variants (Figure 2, upper panel). Similar results were obtained with 0% FBS medium (the condition used for protein production before purification), except for I343V, where its increased amount became comparable to that of other variants exhibiting strong signals (Figure 2, lower panel).

All the recombinant histidine-tagged proteins, produced in serum-free media, were then purified using nickel-affinity chromatography. Regarding R364W variant, its secretion was very low in the supernatant of stable CHO cells; thus, higher amounts of cells were required to obtain enough purified protein in sufficient amount to perform functional experiments. 

### 2.3. Evidencing of the O-Fucosyltransferase Activity for POFUT1 Variants

The *O*-fucosyltransferase activity of WT protein and each POFUT1 variant was assessed in vitro, following incubation with an acceptor substrate (EGF-LD) and a chemically modified donor substrate (GDP-azido-fucose). As previously described [25], click chemistry reaction (CuAAC) coupling alkynyl-biotin to the azido-modified *O*-fucose, followed by blotting techniques using labeled peroxidase-streptavidin, were then performed to reveal fucose transfer to EGF-LDs (Figure 3).

We first tested WT POFUT1 activity with the two NOTCH1 EGF-LDs, namely EGF-LD 12 and EGF-LD 26, known to be modified with *O*-fucose by POFUT1 [10]. Using the same procedure as previously reported, they were produced as isolated proteins in *E. coli* BL21 strain and purified. As expected, they were recognized previously as correctly folded and able to receive *O*-fucose by POFUT1 [25]. As negative controls, we used T466A EGF-LD 12 and T997A EGF-LD 26, having their *O*-fucosylation sites mutated.

As expected, specific intense bands were observed at the proper size for both NOTCH1 WT EGF-LDs 12 and 26 incubated in vitro with human WT POFUT1 while no signal was obtained with their T/A mutated counterparts (Figure 3a). This result confirmed the successful transfer of azido-fucose to threonine residues of both *O*-fucosylation consensus sites of each EGF-LD. Similarly to mouse POFUT1 [25], the human counterpart was able to transfer fucose more efficiently to EGF-LD 26 than to EGF-LD 12 in vitro, since signals were obtained after 1 h and 20 h incubation with POFUT1 respectively. This led us to only perform in vitro *O*-fucosylation assays with EGF-LD 26. All recombinant POFUT1 variants were assessed in the same way with WT EGD-LD 26 and its negative control T997A (Figure 3b). Surprisingly, specific and strong signals were obtained in all cases, meaning that no mutation prevented WT EGF-LD 26 from being specifically modified by *O*-fucose at T^997^. However, signals with variable intensities were clearly observed, strongly suggesting that mutations affected POFUT1 *O*-fucosyltransferase activity. Indeed, all in vitro *O*-fucosylation assays were carried out with equal quantities of EGF-LD and POFUT1 variants, determined following protein quantification using the BCA method.

### 2.4. Quantitative MRM-MS Analysis of the O-Fucosyltransferase Activity for Each POFUT1 Variant

MRM-MS is a highly sensitive method of targeted mass spectrometry (MS) which is used to selectively detect and quantify peptides, obtained after protein reduction, alkylation, and digestion [31]. MRM-MS was performed to quantify the ability of each POFUT1 variant to transfer fucose compared to WT POFUT1 as a reference, in the same way as previously described in Reference [25].

For the in vitro *O*-fucosylation assays, the same amount of each POFUT1 variant was mixed with the same quantities of WT EGF-LD 26 (as a relevant acceptor substrate) and unlabeled GDP-fucose (as the donor substrate). After 20 h of incubation at 37 °C followed by trypsin digestion, the MRM-MS method was performed to confirm our previous results using click chemistry and to quantify the *O*-fucosyltransferase capacity of POFUT1 variants. Spectra obtained showed that human WT POFUT1 was indeed able to transfer specifically *O*-fucose to WT EGF-LD 26 (Figure 4) but with less efficiency (25.00 ± 20.68%) (Figure 5) than mouse recombinant POFUT1 (about 50%) [25]. All POFUT1 variants can partially modify at different degrees EGF-LD 26 with *O*-fucose since both non-modified and *O*-fucosylated peptides were detected. Regarding the mutants R364W, I343V, and D348N, peaks with more intensity were clearly obtained for the *O*-fucosylated forms than for the non-modified ones. This was consistent with an increased ability of these POFUT1 variants to transfer *O*-fucose on this EGF-LD (Figure 4). The variant I343V exhibited the best rate of *O*-fucosylation (86.45 ± 9.59%) (Figure 5), close to the rates obtained for the POFUT1 variants D348N, R364W, and surprisingly T115A. To a lesser extent, the R43H and Y73C mutations also led to a significantly increased *O*-fucosyltransferase activity (Figure 5). Finally, POFUT1 activity was not significantly affected when the non-conserved S^300^ residue was mutated to leucine.

## 3. Discussion

POFUT1 was recently shown to exert an essential role in the colorectal progression from precancerous lesions (adenomas) to carcinoma [32]. The overexpression of this *O*-fucosyltransferase, which is an enzyme involved in the modulation of NOTCH signaling activation, was found in CRC from stage I and was associated with tumor progression and metastasis [6]. On the contrary, its silencing in CRC cells led to inhibition of cell proliferation and diminished cell invasion and migration in vitro and in vivo [7]. Besides, *POFUT1* overexpression was found in many other cancers and notably in hepatocellular carcinomas, where it was associated with a poor prognosis [20]. Overall, these results highlighted an undeniable oncogenic activity of POFUT1 in cancer such as CRC.

*POFUT1* overexpression was mainly associated with chromosomal amplification, leading to an increased amount of POFUT1 [3,6]. However, little is known about the modulation of POFUT1 activity, potentially caused by missense mutations related to human diseases, including CRC. A recent approach was conducted to determine how POFUT1 missense mutations could impact NOTCH1 signaling (R240A, M262T, S356F and R366W) found in Dowling-Degos disease (DDD), an autosomal dominant genodermatosis [30]. Three of these mutations located at highly conserved positions within or close to the substrate-binding regions (R240A, S356F, and R366W) were deleterious for Notch activity. However, the mutation of the residue M^262^ (M262T), which is not conserved in all species had no effect. To date, no other such point mutations were reported for POFUT1 in human diseases.

In this study, structure-function studies were carried out for seven POFUT1 mutated variants (R43H, Y73C, T115A, S300L, I343V, D348N, R364W), resulting from point mutations and found in CRC patients according to BioMuta database. According to information found in this database regarding POFUT1, some of these seven residues (R^43^, T^115^, D^348^) were also mutated in other cancers such as uterine cancer (R43C), melanoma (T115A, D348N), and lung cancer (D348Y). These data could emphasize that these residues, including the less conserved residue (T^115^) among species, are critical for POFUT1 activity and that their mutation could impair its *O*-fucosyltransferase activity.

Using the chemo-enzymatic approach to reveal the *O*-fucosyltransferase activity, we first showed that no mutation found in POFUT1 from CRC patients leads to a loss of enzymatic activity. On the contrary, all POFUT1 variants allowed an efficient azido-fucose transfer to NOTCH1 EGF-LD26. Indeed, the quantification of EGF-LD26 *O*-fucosylation using MRM-MS even highlighted a significant gain-of-function for the mutated POFUT1 variants, except for S300L. This mutation S300L is located in a region with low levels of conservation among species, namely the 299-304 region corresponding to the end of an α-helix of the Rossmann domain followed by a turn, far away from the catalytic center, as shown in Figure 6a.

This study describes, for the first time, some mutated human POFUT1 variants as more active than WT POFUT1. To progress in understanding the effects of mutations on POFUT1 activity, especially those close to the GDP-fucose binding cavity (R43H, Y73C), automatic modelling by docking was performed using the COACH-D Server [33,34]. Possible explanations were proposed based on knowledge of interactions between the complex POFUT1/GDP-fucose with EGF-LD [10]. Since POFUT1 R^43^ residue was known to interact with the fucose moiety of GDP-fucose directly, the meta-server COACH-D was used to predict using molecular docking how the rearrangement of the catalytic cavity of mutated R43H POFUT1 could still allow binding to GDP-fucose without generating steric clashes. Interestingly, the proposed rearrangement (with a confident score of prediction (C-Score) of 0.79) of the binding cavity with a much more deeply buried H^43^ than R^43^ in the WT counterpart might result in a more exposed fucose moiety (Figure 6b,c). This led us to think that this new position of the donor substrate could facilitate fucose transfer to EGF-LD. However, a more complex docking would be required if considering the interaction of three molecules, namely POFUT1, GDP-fucose, and EGF-LD. In addition to a potential effect on GDP-fucose binding, the Y73C mutation is likely to affect EGF-LD binding since the equivalent residue in mice (Y^78^) was known to be involved in the interaction with EGF-LD [10,30]. Indeed, a stacking interaction was highlighted between Y^78^ of POFUT1 and the Gly residue of EGF-LD. This effect was observed frequently at the position C^2^+3 of *O*-fucosylable hEGFs [10]. While Asp or Glu at this position was shown in the latter study to induce a steric clash and to lead to a weakened interaction, a cysteine at this position (as for the mutation Y73C found in CRC) could establish a hydrogen bond with a EGF-LD reinforcing interaction, thereby leading to increased POFUT1 activity.

We noticed that three mutations (I343V, D348N, R364W) leading to the most increased in vitro POFUT1 activity, with similar *O*-fucosylation rates ranging from 83.08 ± 7.17% to 86.45 ± 9.59% (compared to that of WT POFUT1 of 25.00 ± 20.68%), were all located in the highly-conserved region L341-D367 comprising the substrate-binding region S357-F358. Indeed, these three mutations are in the same Rossmann domain, relatively close to the binding pocket of GDP-fucose compared to the mutation T115A. Surprisingly, T115A mutation also gave rise to a significant increase of POFUT1 activity towards NOTCH1 EGF-LD26 while S300L did not have any effect. Indeed, T115 appeared to be less conserved among species than S300. We can hypothesize that mutations leading to a gain-of-function for POFUT1 induced a global conformation change favoring the accessibility of binding pocket to GDP-fucose, thereby leading to a more efficient transfer to EGF-LD 26. Additional experiments should be carried out to understand better the molecular interactions between mutated POFUT1 proteins and different EGF-LDs, whose interaction with WT POFUT1 is known. This could shed light on the gain-of-function induced by missense mutations found in POFUT1 from CRC patients. Recently, an increased amount of WT POFUT1, resulting from a chromosomal amplification of the region 20q11.21, was shown in CRC and correlated to tumor progression through NOTCH signaling activation [6]. A gain-of-function for POFUT1 could have similar effects in CRC patients carrying the missense mutations described in this study. In both cases, protein targets of POFUT1 such as NOTCH receptors and their ligands might be affected by POFUT1 overexpression or hyperactivity. This could have led to more *O*-fucose transferred to these protein partners, influencing receptor-ligand interactions, and subsequently, NOTCH signaling activation. However, Notch receptors and ligands are not the only targets of POFUT1. Indeed, they belong to the 87 human proteins identified to have at least one EGF-LD with a potential *O*-fucosylation consensus sequence [35]. Thus, it is likely that other membrane or secreted EGF-LD-containing proteins, predicted to be modified with *O*-fucose, are affected by overexpressed or hyperactive POFUT1 in CRC. It could be interesting to identify specifically, in CRC samples, the proteins affected by increased expression or activity of this ER-resident *O*-fucosyltransferase 1. The identification of these proteins would be helpful to provide more insights into the contribution of POFUT1 to the tumor process.

In view of this study, the presence of these mutations in new CRC patients could be associated with colorectal tumor aggressiveness. Therefore, these mutations support previous studies [6,22] considering POFUT1 as a potential biomarker for CRC and other cancers.

## 4. Materials and Methods

### 4.1. Plasmid Constructs to Produce Recombinant EGF-LDs and POFUT1 Variants

Plasmid constructs using pET-25b(+) vector (Novagen, Millipore, MA, USA) for bacterial production of WT NOTCH1 EGF-LDs 12 and 26, as well as their counterparts T/A mutated at their *O*-fucosylation site (T466A and T997A, respectively), were previously described in the Appendix A and Methods [25]. The modified vector named pSec-NtermHis6, used in the latter study and derived from the commercial pSecTag/FRT/V5-His-TOPOR vector (Thermo Fisher Scientific, Waltham, MA, USA) was used to express a secreted form of recombinant human POFUT1 with an N-terminal polyhistidine tag. Indeed, human POFUT1 (NP_056167.1) (residues 27-384) cDNA without its signal peptide sequence and its C-terminal KDEL-like motif replaced by a stop codon, was cloned after PCR amplification between *Kpn*I and *Bam*HI sites in the modified vector named pSec-NtermHis6. The resulting plasmid construct referred to as pSec-NtermHis6Pofut1 was used as a template to generate the seven mutated plasmid constructs each carrying a missense mutation found in CRC, using the GENEART^®^ Site-Directed Mutagenesis System (Invitrogen, Carlsbad, USA) according to the manufacturer’s protocol. After nucleotide sequence verification, each plasmid construct was subjected to a cotransfection with pOG44 vector (Thermo Fisher Scientific, Waltham, MA, USA) expressing the Flp recombinase to produce stably transfected Flp-In^TM^ CHO cells (Thermo Fisher Scientific, Waltham, MA, USA).

### 4.2. Cell Culture and Transfection

CHO Flp-In^TM^ cells were cultured in an F12 medium (Thermofisher Scientific, Waltham, MA, USA) supplemented with 10% FBS (Biowest, EuroBio, Courtaboeuf, France) and 0.5% penicillin/streptomycin (Gibco, Carlsbad, CA, USA) and 100 µg×mL^−1^ of Zeocin^TM^ (Thermo Fisher Scientific, Waltham, MA, USA), in a humidified atmosphere with 5% CO_2_. CHO Flp-In^TM^ cells were cotransfected with 1 µg of plasmid DNA and 10 µg of pOG44 vector by lipofection with X-tremeGENE^™^ DNA Transfection Reagent (Sigma-Aldrich, Saint Louis, MO, USA) according to the manufacturer’s protocol. After transfection, the selection was initiated by changing medium with complete F12 medium containing 500 µg × mL^−1^ of Hygromycin B (Thermofisher Scientific, Waltham, MA, USA). Hygromycin B-resistant cells were then amplified and controlled for production of recombinant POFUT1 variants in culture medium by Western blot.

### 4.3. Cell Lines

In this study, we used the *E.coli* BL21 strain to produce recombinant peptides EGF-LDs as described in Reference [25]. We also used Flp-In^TM^ CHO cells (Thermo Fisher Scientific, Waltham, MA, USA) and eight established cell lines corresponding to Flp-In^TM^ CHO cells, stably expressing WT POFUT1 or one of the seven recombinant mutated (R43H, Y73C, T115A, S300L, I343V, D348N, and R364W) POFUT1 variants.

### 4.4. Protein Production and Purification

Recombinant human WT and mutated POFUT1 variants with an N-terminal polyhistidine tag were produced as secreted proteins by stable Flp-In^TM^ CHO cells. After production during 72 h in serum-free F12 medium, proteins were recovered by centrifugation from cell culture supernatants, concentrated in binding buffer (25 mM Tris-HCl, 500 mM NaCl, 5 mM CaCl_2_, 20 mM imidazole, pH 7.5) and purified on a Ni-NTA column by imidazole gradient using AKTA prime system (GE Healthcare, Piscataway, NJ, USA). Recombinant WT and T/A mutated EGF-LDs of mouse NOTCH1 were produced in BL21 and purified as previously described in Reference [25]. All purified recombinant proteins used in this study were concentrated through Amicon ultra centrifugal filters 3K or 10K in Tris-CaCl_2_ (25 mM Tris, 5 m CaCl_2_, pH 7.5) and quantified using a bicinchoninic acid (BCA) protein assay (Sigma-Aldrich, Saint Louis, MO, USA) with bovine serum albumin as a standard.

### 4.5. SDS-PAGE and Blotting Techniques

For POFUT1 variants, purified proteins were resolved by SDS-PAGE using 12% polyacrylamide gels and transferred to 0.45 µm nitrocellulose membrane (GE Healthcare, Buckinghamshire, UK), for 90 min at 0.8 mA per cm^2^. Membranes were blocked with TBS-T (50 mM Tris, 150 mM NaCl, pH 7.6 (TBS), supplemented with 0.1% Tween-20 (*v/v*)) and 5% (*w/v*) fat-free milk, for 1 h at room temperature. Membranes were then incubated with anti-POFUT1 antibody [36], diluted at 1:2000 in TBS with 0.5% Tween-20 (TBS-T_0,5_) containing 5% (*w/v*) of fat free milk overnight at 4 °C. After three washes with TBS-T_0.5_, the membranes were incubated with anti-rabbit HRP-conjugated IgG (Dako, Glostrup, Denmark), diluted at 1:3000 in TBS-T_0.5_ containing 2% (*w/v*) non-fat dry milk for 1 h at room temperature. After three washes in TBS-T_0.5_, proteins were revealed after addition of ECL^TM^ Prime Western blotting detection reagent (GE Healthcare, Uppsala, Sweden) and visualized using an Amersham Imager 600 device (GE Healthcare, Uppsala, Sweden). 

After reactions by click chemistry, EGF-LDs were separated on 15% polyacrylamide gel and transferred to 0.2 µm nitrocellulose (GE Healthcare, Buckinghamshire, UK), for 30 min at 0.8 mA per cm^2^. Membranes were blocked with 10% fat-free milk-TBST for 10 min, before being incubated with streptavidin-HRP in TBS-T_0.5_ at 25 ng.mL^−1^ for 30 min. The membranes were washed three-times before and after streptavidin-HRP incubation with TBST, for 15 min per wash. The membranes were revealed as described above.

### 4.6. Glycosyltransferase Reaction

Before the CuAAC experiments, glycosyltransferase reactions were carried out with 1µg of WT POFUT1 or one mutated POFUT1 variant, mixed with 2 nmoles of GDP-azido-fucose (R&D Systems Inc., Minneapolis, MN, USA) as recommended by the manufacturer (R&D Systems) and 2 µg of purified EGF-LD in 25 µL of reaction buffer (25 mM Tris, 5 mM CaCl_2_, 10 mM MnCl_2_, pH 7.5), and then incubated for 1 h or 20 h at 37 °C.

For mass spectrometry analysis, 1 µg of WT or mutated POFUT1 was incubated with 2 µg of purified EGF-LD and 2 nmoles of GDP-fucose in 20 µl of reaction buffer and incubated for 20 h at 37 °C. 

### 4.7. Click Chemistry Reactions

CuAAC was performed using 1.25 mM CuCl_2_, 2.5 mM ascorbic acid, and 0.125 mM alkynyl biotin (R&D Systems Inc., MN, USA), directly added to the glycosyltransferase reactions. The mixture was incubated in the dark for 1 h at room temperature.

### 4.8. Targeted Mass Spectrometry

Proteins were reduced, alkylated, and digested by Glu-C. Resulting peptides were analyzed with a nanoLC 425 in micro-flow mode (Eksigent, Dubli, CA, USA) coupled to a TTOF5600+ mass spectrometer (SCIEX, Framingham, USA) in Information-Dependent Acquisition mode. ProteinPilot 5.0 (SCIEX) was applied to search against the recombinant protein sequence database, and the MRM transition list was established using Skyline 3.5.0 (MacCoss Lab, University of Washington, Seattle, WA, USA). The *O*-fucose was added in silico at the expected position with PeakView software (SCIEX, Framingham, MA, USA), and *m/z* of precursor and fragments were calculated. The data were acquired in high-Resolution MRM mode and processed with MultiQuant Software 3.0.1 (SCIEX, Framingham, MA, USA). Areas were collected for the same most intense fragment of *O*-fucosylated and non-modified peptides, and a ratio of *O*-fucosylation was calculated.

### 4.9. Alignments and Automatic Molecular Modelling

Alignments of POFUT1 protein sequences were performed with FASTA sequences of POFUT1 from different species, using the Multalin server (http://multalin.toulouse.inra.fr) [37]. Using MatchMaker of UCSF CHIMERA (http://www.rbvi.ucsf.edu/chimera) (Resource or Biocomputing, Visualization, and Informatics at the University of California, San Francisco, CA, USA, with support from NIH P41-GM103311) [38], mouse NOTCH1 EGF-LD 26, co-crystallized with mouse POFUT1 (PDB 5KY4) [10], was superimposed with co-crystallized human POFUT1 with GDP-fucose (PDB 5UXH) [30]. Docking experiments were made using the COACH-D server (https://yanglab.nankai.edu.cn/COACH-D/), FASTA sequences of WT or R43H POFUT1 variant, and GDP-fucose (in sdf format file) as a donor substrate. The proposed model for R43H POFUT1 interacting with GDP-Fucose as a substrate was selected according the best C-score, in Pose^U^, and visualized using UCSF CHIMERA.

### 4.10. Statistical Analyses

All the experiments were performed at least three times independently. Statistical comparisons were achieved using the t-Student test implemented in GraphPad Prism 7 (GraphPad Software Inc, San Diego, CA, USA). Results were considered statistically significant if the *p*-value was less than 0.05.

## 5. Conclusions

In conclusion, our findings indicate that among the rare missense mutations affecting POFUT1 and found in patients with colorectal cancer, six of them induced an increase of *O*-fucosyltransferase activity compared to WT POFUT1. These mutated variants could modify the *O*-fucosylation status of POFUT1 protein targets such as NOTCH1 receptor and its ligands, and subsequently promote colorectal cancer.

## Figures and Tables

**Figure 1 cancers-12-01430-f001:**
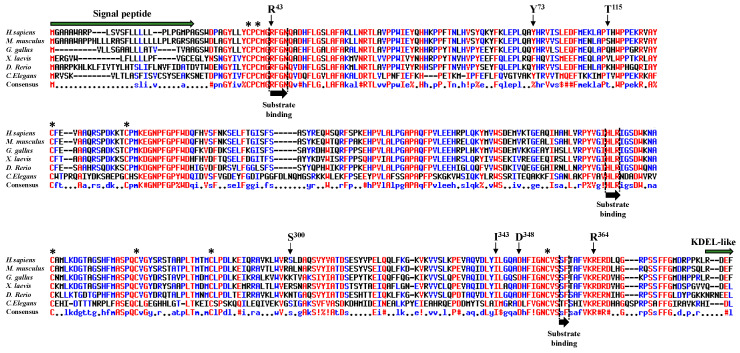
Alignment of POFUT1 protein sequences. Using the Multalin server, multiple sequence alignment for POFUT1 was done for different species such as *Homo sapiens*, *Mus musculus*, *Gallus gallus*, *Xenopus laevis*, *Danio rerio,* and *Caenorhabditis elegans*. Among the seven missense mutations found in POFUT1 from CRC patients, five of them corresponded to mutations of highly conserved residues (R^43^, Y^73^, I^343^, D^348^, and R^364^), even located in a substrate-binding region for some residues such as R^43^. However, POFUT1 T^115^ and S^300^, mutated in CRC patients, were not conserved among species. The three conserved regions known to be involved in binding to GDP-fucose are indicated with black arrows and specific sequences (signal peptide and KDEL-like ER-retention signal) with green arrows. Asterisks indicate the positions of conserved cysteines forming disulfide bridges.

**Figure 2 cancers-12-01430-f002:**
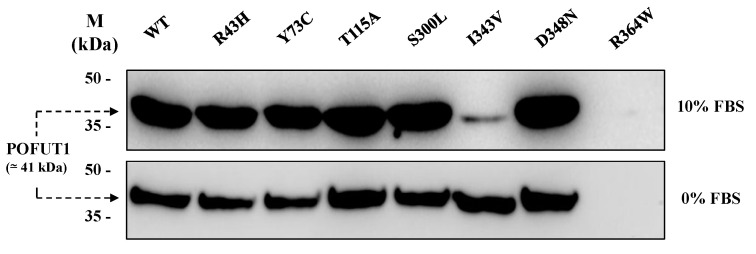
Western blot analysis, using anti-POFUT1 antibody, of recombinant POFUT1 variants secreted by stable CHO cell lines. Supernatants from each stable cell line were recovered by centrifugation from culture media after three days of protein production in F12 media with 10% (upper panel) or 0% (lower panel) fetal bovine serum (FBS). Same volumes of crude supernatants were loaded for WT and mutated POFUT1 proteins. Uncropped blots are shown in Appendix A.

**Figure 3 cancers-12-01430-f003:**
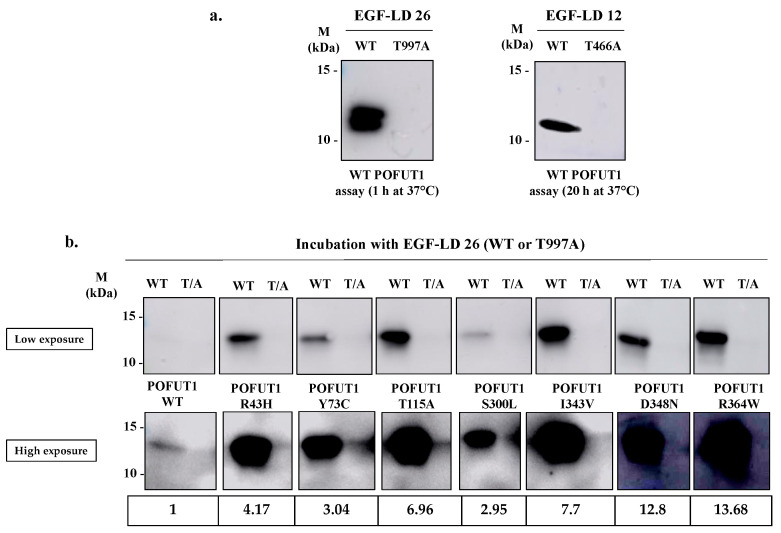
In vitro *O*-fucosyltransferase assay for activity of WT and mutated POFUT1 variants using click chemistry. (**a**) After 1 h or 20 h incubation at 37 °C of recombinant WT human POFUT1 with NOTCH1 EGF-LD 26 (WT and T997A) or EGF-LD 12 (WT and T466A) and GDP-azido-fucose, click chemistry (CuAAC) with alkynyl biotin was performed to link biotin to transferred *O*-linked fucose covalently. After SDS-PAGE and blotting techniques, EGF-LDs modified with *O*-fucose were revealed using HRP-streptavidin. (**b**) After 1 h independent incubations of WT POFUT1 and each mutated POFUT1 variant with WT or T997A EGF-LD 26 at 37 °C, the same procedure as described in (a) was carried out. A low exposure did not allow us to visualize WT POFUT1 labelling. After high exposure, the signal for WT POFUT1 was detected, and quantifications were performed for each POFUT1 variant compared to WT. Uncropped blots are shown in Appendix A.

**Figure 4 cancers-12-01430-f004:**
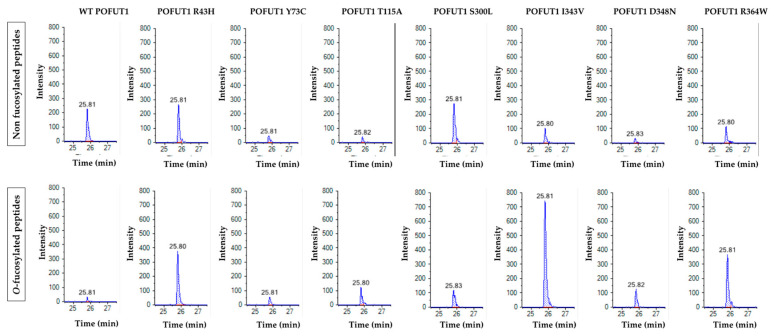
Multiple reaction monitoring mass spectrometry (MRM-MS) analyses of trypsin-digested WT EGF-LD26, after incubations with GDP-fucose and either WT POFUT1 or mutated POFUT1 variants independently. Peaks corresponding to non-modified peptides (upper panels) and peptides modified with *O*-fucose (lower panels) were shown for WT POFUT1 and all POFUT1 variants. Peaks correspond to the most intense MS2 fragment signal, obtained during the run time.

**Figure 5 cancers-12-01430-f005:**
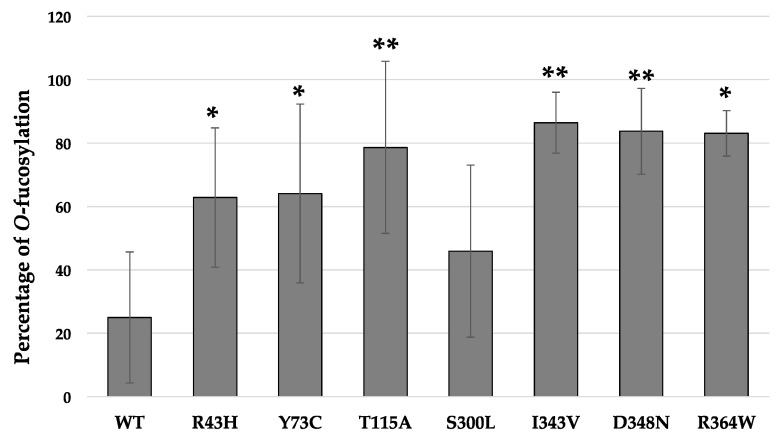
Quantification of the *O*-fucosyltransferase activity of WT POFUT1 and its seven mutated counterparts found in CRC patients. The percentage of *O*-fucosylation was determined from previous MRM-MS analyses based on ratios of peak areas of *O*-fucosylated peptides reported to those of total peptides. Bar graph represented mean of percentage ± SD. Each experiment was performed three times independently. Statistical significance was assessed using a two-tailed Student test; * *p* < 0.05, ** *p* < 0.01 vs. WT.

**Figure 6 cancers-12-01430-f006:**
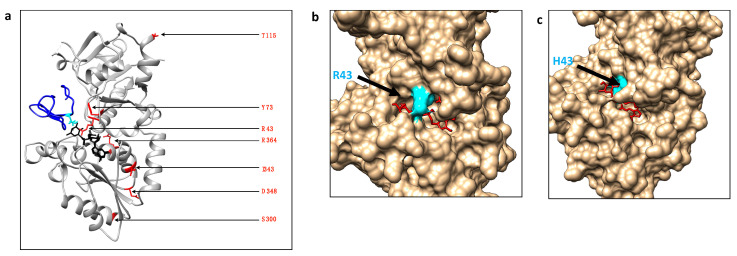
Location of POFUT1 residues mutated in CRC and automatic docking models for WT POFUT1 and R43H variant with GDP-fucose as a ligand. (**a**) Using Matchmaker of CHIMERA, X-ray structure of human POFUT1 (grey) co-crystallized with GDP-fucose (black) (PDB 5UXH) was superimposed with mouse POFUT1 co-crystallized with EGF-LD 26 (blue) (PDB 5KY4). After removal of mouse POFUT1, residues implicated in CRC mutations were shown with their sidechains (red). (**b**) Using the COACH-D docking server, we performed automatic molecular docking using the FASTA sequence of WT human POFUT1 as a template and GDP-fucose (in red) as a ligand (donor substrate). The residue R^43^, located in front of the binding pocket to GDP-fucose, was colored in cyan. (**c**) In the same way, a structural model was generated for mutated POFUT1 in which the H43 moved to a more deeply buried position in the binding pocket of GDP-fucose, compared to R43 in WT POFUT1. This could lead to a more exposed fucose moiety of the donor substrate to improve fucose transfer to an EGF-LD.

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
