# Peer review of "Functional Characterization of POFUT1 Variants Associated with Colorectal Cancer"

_cancers, 2020, doi:10.3390/cancers12061430_

Round 1

Reviewer 1 Report

In this manuscript the authors have investigated the functional effects of non-synonymous point mutations within the coding region of POFUT1. The product of this gene is an O-fucosyltransferase which has previously been found to be overexpressed in some cancers such as CRC, dysregulating the Notch pathway. The authors have tested 7 mutations to investigate if they have loss or gain of function effects using in vitro methods. Their methods involve the generating recombinant proteins and the use of mass spectrometry to quantify enzyme activity on substrates derived from NOTCH, and have mostly been previously published. In the main finding of this study, the authors report that 6 out of the 7 mutations increase O-fucosyltransferase activity above that seen in their wildtype. This main finding is quite convincing and the role of POFUT1 in CRC is of interest to the field.

However the data in this paper is not always well presented.

  1. In figure 3 the wildtype blot is separated from those of the mutant proteins. These should be performed on the same gel. Some quantification of the figure 3 bands (such as that shown in sup p figure 2 for western blot analysis) would be good.
  2. Figure 4 is the main finding of the paper and only focuses on the most active mutant. The other data is shown in supp figure 1 and the table. This data could be better presented. Either all the panels should be shown in the main text or the table could be converted to graphical format or both. Error bars should be added to show variation between replicates.

Minor points:

The section describing Figure 5 would be better placed in the results rather than discussion. Could the highest activity variants also be modelled in this way?

The Click chemistry is a specialist technique and should be described in more detail in the methods.

A short section should be added to the discussion about how this research could impact cancer patients in the clinic. e.g. could POFUT1 expression be used as a biomarker, or a therapeutic target etc

Reviewer 2 Report

Summary: Alterations in POFUT1, including overexpression, is observed in CRC and other cancers.  This altered expression can affect Notch signaling, influencing tumorigenesis.  Therefore, interrogation of POFUT1 expression, including mutations of this gene, is important.  The authors note that, infrequently, missense mutations are observed with the POFUT1 gene, instead of overexpression.  Seven missense mutations were identified. Recombinant forms of these were produced and analyzed for their O fucosyltransferase ability compared to the wild-type.  Six out of the seven variants demonstrated increased activity. It is therefore possible that these variants mimic overexpression in their ability to modulate the activity of Notch (and other) signaling.  This is novel in that it is the first demonstration of a gain of function mutation in POFUT1, with implications for CRC (and other cancers). 

General:Strengths – the methodology was mostly sound, this was a solid mechanistic study, and the findings were well demonstrated and the interpretations reasonable.  The authors essentially accomplished what they set out to do for this study.

Weaknesses – there are several problems with the writing and the statistics, as noted in the specific critiques. As regards the protein blotting, is there any way of normalizing other than equal volumes?  One major weakness in the study is the lack of definitive downstream functional effects demonstrated.  For example, demonstration that Notch signaling is indeed deregulated as a function of the production of the mutant POFUT1 proteins, such as the use of Notch reporter assays (or some other method).  I realize that a paper cannot contain all relevant experiments, at some point one study must end and future work begins subsequently.  However, given that some of these additional experiments would not be particularly technologically onerous, do the authors have justification for not showing actual effects on Notch (or other) signaling?  

Specific criticisms: Grammar and punctuation problems, line 74 and lines 148-9.  Further, the sentence of lines 129-130 is awkwardly constructed. It would be helpful as well to specifically state the problem referred to in that sentence.  In data such as Fig. 2, can some normalization be performed, such as ponceau staining (at least for the 0% FBS) or to contrast to total protein (e.g., ponceau) form the cell lysates from the equivalent number of cells used to generate each secreted fraction?  Admittedly, these are imperfect normalization methods; what level of confidence do the authors have that the data they present represent quantitative data?  And if the experiments were duplicated, are the botting data shown representative of all replicates? Table I utilizes SEM.  That is inappropriate, SD should be used; SEM is not a descriptive statistic and the statistical evaluation is variation of the data points (within a single study), not the precision of the means.  The lack of functional assays is for the entire manuscript, not any specific line, figure, or table.
